# The Prediction of Residual Electrical Life in Alternating Current Circuit Breakers Based on Savitzky-Golay-Long Short-Term

**DOI:** 10.3390/s23156860

**Published:** 2023-08-01

**Authors:** Junfeng Ouyang, Changchun Chi

**Affiliations:** School of Electrical Engineering, Shanghai Dianji University, Shanghai 200240, China; oyjf845739813@163.com

**Keywords:** AC circuit breaker, principal component analysis, maximum information coefficient, Savitzky–Golay convolutional smoothing algorithm, long short-term memory neural network

## Abstract

In order to improve the accuracy of predicting the remaining electrical life of AC circuit breakers, ensure the safe operation of electrical equipment, and reduce economic losses caused by equipment failures, this paper studies a method based on the Savitzky–Golay convolution smoothing long short-term memory neural network for predicting the electrical life of AC circuit breakers. First, a full lifespan test is conducted to obtain degradation data throughout the entire life cycle of the AC circuit breaker, from which feature parameters that effectively reflect its operational state are extracted. Next, principal component analysis and the maximum information coefficient are used to remove redundancy in the feature parameters and choose the best subset of features. Subsequently, the Savitzky–Golay convolutional smoothing algorithm is employed to smooth the feature sequence, reducing the impact of noise and outliers on the feature sequence while preserving its main trends. Then, a secondary feature extraction is performed on the smoothed feature subset to obtain the optimal secondary feature subset. Finally, the remaining electrical lifespan of the AC circuit breaker is treated as a long-term sequence problem and the long short-term memory neural network method is used for precise time-series forecasting. The proposed model outperforms backpropagation neural networks and the gate recurrent unit in terms of prediction precision, achieving an impressive 97.4% accuracy. This demonstrates the feasibility of using time-series forecasting for predicting the residual electrical lifespan of electrical equipment and provides a reference for optimizing the method of predicting remaining electrical life.

## 1. Introduction

When it comes to the safety and reliability of power systems, low-voltage AC circuit breakers are considered essential protective devices. They are widely used in transmission, distribution, and generation fields. Their primary function is to rapidly disconnect faulty circuits in the event of a fault, ensuring the safe operation of the power system [1]. The remaining electrical life of an AC circuit breaker reflects its breaking performance in the circuit. Accurately predicting the remaining electrical life of an AC circuit breaker can detect potential faults in a timely manner, thereby improving the reliability of the power system and reducing possible economic losses.

Therefore, analyzing the aging mechanism of circuit breakers and studying the changing trends of characteristic parameters are of significant importance for evaluating the electrical lifespan of circuit breakers, as well as assessing the performance and reliability of the entire power system [2]. By combining experimental data with mathematical models, it is possible to predict the remaining electrical lifespan of circuit breakers, providing a basis for maintenance and ensuring the normal operation of power systems.

For the prediction of the remaining electrical lifespan of low-voltage circuit breakers, commonly used approaches include those based on physical models, experimental data, and intelligent algorithms [3,4]. Physical model-based methods involve an in-depth study of the internal physical mechanisms and failure modes of circuit breakers, establishing mathematical prediction methods. Their advantages lie in detailed system modeling and parameter analysis, along with high interpretability. Experimental data-based methods utilize accumulated operational data of circuit breakers to perform statistical analysis on failure probabilities, thereby obtaining predictions of the remaining lifespan. On the other hand, intelligent algorithm-based methods rely on techniques such as machine learning and neural networks to build prediction models by learning and analyzing large amounts of data, enabling the prediction of the circuit breaker lifespan. It should be noted that the selection of appropriate methods and techniques for feature extraction and the prediction of the remaining electrical lifespan in low-voltage circuit breakers depends on practical situations and application requirements. Continuous optimization and improvement should be carried out during the utilization process. Feature extraction methods and remaining lifespan prediction methods for low-voltage circuit breakers represent important research directions in power systems. Many scholars have already proposed various methods and algorithms in this field.

F. Xing et al. improved the prediction model by considering the influence of arc duration on contact erosion, resulting in increased accuracy [5]. Liu Z et al. studied the impact of arc voltage on the electrical lifespan of circuit breakers, using Monte Carlo methods to simulate the distribution of electrical lifespan under different operations [6]. K. Huang et al. proposed a fuzzy reliability-based lifespan assessment model and an improved gray prediction method based on stepwise data sequences to predict the remaining lifespan of circuit breakers [7]. W. De et al. used gray system theory and LabVIEW programming to develop a prediction model for circuit breaker lifespan [8]. K. Li established a prediction model for electrical remaining lifespan based on the feature parameter of contact loss, assuming a degradation process following the Wiener process [9]. Li K et al. fitted cumulative arc erosion frequency empirical distribution functions and histograms with gamma cumulative curves and probability density curves, demonstrating that the degradation process of AC contactors follows a gamma process, and performed remaining lifespan prediction using the gamma process model [10]. S. Sun et al. selected contact voltage during closing and arc energy during opening as feature parameters, calculated their correlation coefficients with electrical lifespan using Spearman rank correlation, and then combined the contact voltage and arc energy proportionally to establish a univariate regression lifespan prediction model [11]. W Zeng conducted in-depth research on the degradation process of relays and established a relay performance degradation model based on linear regression theory, using the model to predict the lifespan of aerospace relays [12]. Junqiang Liu pointed out that the degradation process exhibits multi-stage characteristics due to uncertainty and proposed a multi-stage performance degradation model based on the Wiener process for aircraft engines, along with a method to predict the remaining lifespan [13].

Overall, these studies have made valuable contributions to the prediction of AC circuit breaker remaining electrical lifespan. However, there are still challenges to address, such as the need for a comprehensive representation of degradation information using a larger set of feature parameters, dealing with the nonlinear and non-smooth characteristics of electrical monitoring signals, and effectively utilizing the relationship between current and historical states to improve prediction accuracy.

With the advancement of artificial intelligence technology, the use of deep learning [14] and machine learning [15] in the field of electrical engineering is becoming increasingly widespread. Deep learning techniques have emerged as a powerful tool for building and analyzing models using deterioration data from electrical equipment. This eliminates the necessity for precise degradation statistical models and physical models, effectively addressing the challenge of estimating the remaining electrical life of complex electrical equipment.

Z.HE and colleagues used the arc voltage and arc current signals gathered during the life testing process to extract features that reflect the degradation process of miniature circuit breakers. These extracted features were used as input parameters to construct a life assessment model for miniature circuit breakers using the backpropagation neural network (BPNN). The research results indicated that the extracted relative closing time and drop-off time could significantly reflect the deterioration process of miniature circuit breakers. The experimental data testing showed that the life assessment model constructed using trend characteristic quantities achieved good evaluation results [16]. S.Sun et al. proposed a deep learning-based method for predicting the remaining life of low-voltage circuit breakers. The method involves extracting vibration segments that represent the mechanical performance of the contact system and constructing a multi-channel convolutional autoencoder long short-term memory network (MCCAE-LSTM) as the prediction model. The MCCAE captures degradation information from the time-series data, while the LSTM component quantitatively predicts the remaining mechanical life. The combination of MCCAE and LSTM enables accurate predictions and reduces the impact of data uncertainty caused by system complexity [17]. Su.JZ explored an algorithm that combines the Savitzky–Golay convolution smoothing technique with the BP neural network. This algorithm uses arc duration, arc energy, and arc phase angle as input parameters for the prediction model, allowing for the creation of a contact residual electrical life prediction model [18]. FU.H used wavelet methods to extract time-frequency domain fault-related features and constructed a CNN-LSTM network prediction model based on the extracted features [19]. Sun.SG proposed a residual life prediction algorithm based on the Modified Multi-scale Permutation Entropy and dual attention long short-term memory mechanism. The feature and time attention mechanisms assigned weights to the features and time steps, respectively, allowing for quantitative prediction of the residual life [20].

This study examined a residual electrical life prediction model for low-voltage AC circuit breakers using the SG-LSTM method. The model considers the circuit breaker’s life status as a long-term degradation sequence and takes into account the correlation between multiple feature parameters over time. First, to effectively capture the degradation information of AC circuit breakers, PCA and MIC methods were employed to select an optimal feature subset. Second, the SG convolution smoothing algorithm was applied to smooth the feature sequence and reduce the influence of noise and outliers. Finally, the LSTM model was utilized to consider the correlation between previous and subsequent states, as well as to memorize long-term historical feature information, enabling the prediction of the time series for circuit breakers. Through case studies, the proposed prediction method showed high precision and offered a novel approach for predicting the residual electrical life of switching devices.

## 2. Principles

The theoretical approaches used for this study include principal component analysis (PCA), maximum information coefficient (MIC), Savitzky–Golay convolution smoothing computational method, and LSTM.

### 2.1. Principal Component Analysis

PCA [21] is a feature selection method based on the computation of the data covariance matrix. It transforms the original features through a linear transformation into a new set of mutually independent features, preserving the main information and variations in the data. These features are arranged in descending order of variance, allowing for dimensionality reduction and feature selection by choosing the desired number of features. The calculation process for PCA feature selection is as follows:

(1) Let *X* be the sample matrix consisting of all samples in the dataset, represented as *X* = [$x_{1},x_{2},\ldots ,x_{m}$]. The overall sample mean is calculated as follows:(1)$$\overset{-}{x}=\frac{1}{m}\sum_{i=1}^{n}x_{i}$$

The covariance matrix of the sample matrix can be obtained from the mean values as follows:(2)$$C=\left(X-\overset{-}{x}\right){\left(X-\overset{-}{x}\right)}^{T}$$

(2) Calculate the eigenvalues $\lambda_{i}$ and eigenvectors of the sample covariance matrix. Calculate the contribution rate of covariance based on the eigenvalues. The contribution rate of the i-th column vector of the sample covariance matrix C and the cumulative contribution rate of the first j columns of the matrix are, respectively,
(3)$$\theta_{i}=\frac{\lambda_{i}}{\sum_{i=1}^{k}\lambda_{i}}$$
(4)$$\Theta_{j}=\sum_{i=1}^{j}\theta_{i}$$

(3) Arrange the feature vectors in a matrix by arranging them in rows from top to bottom according to their corresponding eigenvalues. Set the desired dimensionality reduction to k and take the top k rows to form the matrix P.

(4) Y represents the reduced-dimensional dataset, and the formula for obtaining it is
(5)Y=PX

### 2.2. Maximum Information Coefficient

The maximum information coefficient (MIC), as introduced by D.N. Reshef and his team in 2011, is a measure of correlation between variables based on the theory of mutual information (MI) [22]. Mutual information is a metric that quantifies the similarity of information between two random variables, U and V. The MIC calculation relies on both the marginal and joint probability densities of the variables in question. A higher MI value indicates a higher level of similarity between U and V. MI is calculated using entropy. The entropy of a variable U with probability distribution p(U=u) can be expressed using the following formula:(6)$$H\left(U\right)=-\sum_{x}p\left(U=u\right){log}_{2}^{p}\left(U=u\right)$$

The mutual information I(U,V) between two variables U and V is defined as follows:(7)$$I\left(U,V\right)=\sum_{u\in U}\sum_{v\in V}p\left(u,v\right)log\frac{p(u,v)}{p(u)p(v)}$$
where p(U, V) is the joint probability density of the two random variables X and Y, and p(U) and p(V) are the marginal probability distributions of U and V, respectively.

The MIC calculation involves partitioning the scatter plot of a dataset D into several sub-regions and computing the mutual information for each sub-region between the two variables U and V. The highest mutual information value among these sub-regions is then chosen as the MIC value for the variables. The calculation formula for MIC is as follows:(8)$$MIC\left(D\right)=\max_{\mathrm{UV}<B(n)}{M(D)}_{U,V}=\max_{\mathrm{UV}<B(n)}\frac{I(D,U,V)}{log\left(min(U,V)\right)}$$

Here, “n” represents the sample size, and “B(n)” represents the maximum number of bins or grid cells. Typically, B(n) is chosen as $B\left(n\right)=n^{0.6}$.

### 2.3. Savitzky–Golay Convolution Smoothing Algorithm

The Savitzky–Golay filter (SG) is a time-domain filtering method that effectively removes noise from waveform signals while preserving their shape and width. It achieves this by employing local polynomial least squares fitting. Due to its ability to smooth and denoise data, the Savitzky–Golay filter has found extensive applications in various fields [23].

The effectiveness of data smoothing depends on the chosen window width, which determines the number of data points used to calculate the average value during the smoothing process. The smoothing process can be represented by the following equation:(9)$$z_{k,smooth}=\overset{-}{z_{k}}=\frac{1}{H}\sum_{i=-w}^{+w}z_{k+i}h_{i}$$

To minimize the overall impact of the algorithm on the waveform, each point’s value is multiplied by a smoothing coefficient $h_{i}$. To address the limitations of traditional smoothing algorithms, the least squares principle and polynomial fitting are used to improve the drawback of $\frac{h_{i}}{H}$ in the algorithm.

Suppose the smoothing window has a size of n = 2m + 1, and each point’s value is represented by z = (−m, −m + 2, ..., 0, 1, m − 1, m). The points within the window are fitted using a k − 1 degree polynomial, as follows:(10)$$s=a_{0}+a_{1}x+a_{2}x^{2}+\ldots +a_{k-1}x^{k-1}$$

By applying the least squares method, we can obtain a system of k linear equations. When the number of data points (n) is greater than the number of unknowns (k), the equations will have a solution. Consequently, the fitting parameters A can be determined. Hence, the following relationship holds:(11)$$\left(\begin{array}{c} \begin{array}{c} y_{-m}\\ y_{-m-1} \end{array}\\ \vdots \\ y_{m} \end{array}\right)=\left(\begin{array}{c} 1\\ \;\;\;\;1\;\;\;\;\\ \vdots \\ 1 \end{array}\begin{array}{c} -m\\ -m+1\\ \vdots \\ m \end{array}\begin{array}{c} \;\;\;\cdots \\ \;\;\;\cdots \\ \;\;\;\vdots \\ \;\;\;\cdots \end{array}\begin{array}{c} {\;\;\;(-m)}^{k-1}\\ {\;\;\;\;(-m+1)}^{k-1}\\ \vdots \\ {\left(m\right)}^{k-1} \end{array}\right)\left(\begin{array}{c} a_{0}\\ a_{1}\\ \vdots \\ a_{k-1} \end{array}\right)+\left(\begin{array}{c} e_{-m}\\ e_{-m-1}\\ \vdots \\ e_{m} \end{array}\right)$$

It can be described using the following matrix:(12)$$D_{(2m+1)\times 1}=Z_{(2m+1)\times k}•A_{k\times 1}+E_{(2m+1)\times 1}$$

The least squares solution for matrix A, denoted as $\hat{A}$, is given by
(13)$$\hat{A}={\left(Z^{T}•Z\right)}^{-1}Z^{T}D$$

The smoothing matrix for matrix D is
(14)$$\hat{D}=Z•A=Z•{\left(Z^{T}•Z\right)}^{-1}•Z^{T}•D$$

The “•” in formulas 12 to 14 represents the dot product operation between two matrices.

### 2.4. Long Short-Term Memory Neural Network

The long short-term memory (LSTM) is a type of recurrent neural network used to address the vanishing gradient problem in traditional recurrent neural networks. It is commonly employed for handling time-series data with long time intervals and delays [24]. The internal structure of an LSTM model is depicted in Figure 1. By utilizing different modules within the network, LSTM can assess the importance of current system information and determine whether to retain or forget this information. Hence, LSTM is an effective approach for tackling long-term dependency issues. The structural components of an LSTM unit include an input gate $\left(i_{t}\right)$, a forget gate $\left(f_{t}\right)$, an output gate $\left(o_{t}\right)$, and a self-connected memory cell state $\left(c_{t}\right)$.

In the LSTM network, the forget gate determines if the quantity of information from the previous time step’s cell state $C_{t-1}$ should be preserved in the current moment step’s cell state $C_{t}$. The input to the forget gate is composed of the previous moment step’s hidden state $h_{t-1}$ and the current moment step’s input $x_{t}$. By using a sigmoid function, a decision vector $f_{t}$ is generated, which determines the extent to which information from the previous moment step’s cell state $C_{t-1}$ should be forgotten. The result of the forget gate is obtained by element-wise multiplication between the decision vector $f_{t}$ and the previous moment step’s cell state $C_{t-1}$. The specific calculation formula is as follows:(15)$$f_{t}=\sigma \left(W_{f}\left[h_{t-1},x_{t}\right]+b_{f}\right)$$

By the aforementioned calculation, the forget gate can determine which information from the previous time step’s cell state should be forgotten based on the previous hidden state and the current input. This mechanism enables LSTM to flexibly retain and forget relevant information when processing time sequences, thus capturing the dependencies between sequences more effectively.

The input gate is responsible for determining how much new information should be added to the current state. By using the tanh function, candidate information ${\tilde{C}}_{t}$ is generated. Then, the decision vector $i_{t}$, generated by the sigmoid function, determines how much content from the candidate information ${\tilde{C}}_{t}$ can be added to the cell state $C_{t}$. The specific calculation formula is as follows:(16)$$i_{t}=\sigma \left(W_{i}\left[h_{t-1},x_{t}\right]+b_{i}\right)$$
(17)$${\tilde{C}}_{t}=tanh\left(W_{C}\left[h_{t-1},x_{t}\right]+b_{c}\right)$$

By the aforementioned calculation, it is possible to determine the importance of new information based on the input gate. The candidate information is then multiplied by the decision vector to obtain the updated memory state $C_{t}$. This mechanism allows LSTM to update the memory state based on the current input and the previous hidden state, thereby capturing long-term dependencies in time sequences more effectively.

The output gate decides the output of the current cell and the hidden state that will be conveyed to the next cell. The input of the output gate consists of the previous hidden state $h_{t-1}$ and the present input $x_{t}$. By using the sigmoid function, a vector $o_{t}$ is generated, which determines how much information should be output from the current cell state $C_{t}$. Additionally, the present cell state $C_{t}$ is subjected to the tanh activation function to generate another variable, and these two variables are multiplied to obtain the hidden state $h_{t}$ at the current time step. The specific calculation expression is as follows:(18)$$o_{t}=\sigma \left(W_{o}\left[h_{t-1},x_{t}\right]+b_{o}\right)$$
(19)$$h_{t}=o_{t}tanh\left(C_{t}\right)$$

By the aforementioned calculation, the output gate determines the output of the current cell state and also determines the hidden state that will be passed to the next cell. This mechanism allows LSTM to control the output and transmission of information based on the importance of the current state, enabling it to adapt better to different time-series tasks.

Both the input gate and the forget gate contribute to the cell state $C_{t}$. The cell state $C_{t}$ is obtained by adding the product of the previous cell state $C_{t-1}$ and the forget gate decision vector $f_{t}$, and the product of the input gate decision vector $i_{t}$ and the candidate information ${\tilde{C}}_{t}$. The specific expression is as follows:(20)$$C_{t}=f_{t}*C_{t-1}+i_{t}*{\tilde{C}}_{t}$$

In the equation, $W_{f}$$W_{i}$$W_{C}$ and $W_{o}$ are weight matrices between different layers, while $b_{f}$$b_{i}$$b_{c}$ and $b_{o}$ are the corresponding bias vectors. σ represents the sigmoid activation function.

The recurrent structure of LSTM, as shown in Figure 2, consists of multiple LSTM modules. Through the connections and interactions between these modules, LSTM can effectively handle long-term sequence prediction tasks.

In this structure, each LSTM module consists of an input gate, a forget gate, an output gate, and a cell state. Each module performs calculations based on the current input and the previous hidden state, and then outputs the current hidden state and cell state. By connecting multiple LSTM modules in layers, effective modeling and prediction of long-term sequences can be achieved.

The recurrent structure of LSTM allows for the transmission and interaction of information in the temporal dimension. Through gate mechanisms, the flow of information is controlled, enabling LSTM to effectively capture long-term dependencies in time-series data.

## 3. Prediction Model

### 3.1. Summary of the Prediction Model

This study investigates a residual life prediction model based on an AC circuit breaker with SG-LSTM. Figure 3 illustrates the structure of this model.

The first step involves extracting feature parameters from the voltage and current data of the AC circuit breaker, which reflect its performance degradation state. Subsequently, the importance of these feature parameters for residual life prediction is determined using the principal component analysis (PCA) method. Additionally, the maximum information coefficient (MIC) is used to measure the correlation between feature parameters, eliminating those with low correlation and redundancy with residual life. This results in an optimal set of feature parameters. Next, the SG convolutional smoothing algorithm is employed to smooth the feature sequences, obtaining more stable feature data. Then, secondary feature extraction is performed on the optimal set of feature parameters to capture higher-order relationships and nonlinear patterns among features, yielding the optimal set of secondary feature parameters. Subsequently, the optimal set of secondary feature parameters is subjected to dimensionless normalization to eliminate dimensional differences among different features, ensuring they have the same scale for better application in model training. The preprocessed multidimensional time series is fed into an LSTM neural network, where the structure of the LSTM model is defined based on the specific application, including the number of layers, hidden units, activation functions, etc. The LSTM network extracts temporal information from the sequential data. The remaining life of the AC circuit breaker serves as the prediction label, and the time step and data partitioning are set according to requirements. The LSTM neural network is trained by updating weights and biases iteratively to learn and capture the correlations between the AC circuit breaker’s life cycle and the corresponding data. The model generates predictions by feeding the outputs through a densely connected layer. The resulting predicted values are then transformed through an inverse normalization process to produce the final estimate of the residual life. Finally, evaluation metrics are used to assess and analyze the prediction results, measuring the predictive performance and accuracy of the model.

### 3.2. The Model Loss Function and Evaluation Metrics

In this study, the Adam optimizer is used during the training process to fine-tune the prediction model’s parameters. This optimization technique helps improve the effectiveness of the loss function optimization and enhances the convergence of the model [25]. In this study, the model’s performance is evaluated using the mean square error (MSE) loss function, which calculates the difference between the predicted and actual values. The formula for MSE is as follows:(21)$$Loss=\frac{1}{n}\sum_{t=1}^{n}{\left(y_{t}-{\tilde{y}}_{t}\right)}^{2}$$

In the given equation, n denotes the total number of data points. The symbol $y_{t}$ represents the true value, while ${\tilde{y}}_{t}$ stands for the estimated value.

In addition to the MSE loss function, this study also employs several other evaluation metrics to assess the quality of the predictions, including the root mean square error (RMSE), mean absolute error (MAE), and coefficient of determination ($R^{2}$) as evaluation metrics. RMSE reflects the deviation between the predicted values and the actual values, where a smaller value indicates better proximity to the true values. MAE measures the average magnitude of prediction errors, where a smaller value indicates higher prediction accuracy. The $R^{2}$ metric measures the proportion of variance in the dependent variable that can be explained by the independent variables. A value closer to 1 indicates that the model is better able to capture the relationship between the independent and dependent variables. The formulas are as follows:(22)$$RMSE=\sqrt{\frac{1}{n}\sum_{t=1}^{n}{\left({\overset{-}{y}}_{t}-y_{t}\right)}^{2}}$$
(23)$$MAE=\frac{1}{n}\sum_{t=1}^{n}\left|{\overset{-}{y}}_{t}-y_{t}\right|$$
(24)$$R^{2}=1-\frac{\sum_{t=1}^{n}{\left(y_{t}-{\overset{-}{y}}_{t}\right)}^{2}}{\sum_{t=1}^{n}{\left(y_{t}-\overset{-}{y}\right)}^{2}}$$

In the equation, ${\overset{-}{y}}_{t}$ represents the average value of the actual values.

Therefore, by evaluating the RMSE, MAE, and $R^{2}$, we can comprehensively assess the accuracy of the prediction results, the magnitude of errors, and the suitability of the model to the actual data. Smaller RMSE and MAE values, as well as $R^{2}$ values close to 1, indicate better prediction performance and model fit.

Through the use of the Adam optimizer, MSE loss function, and the selection of evaluation metrics, this study effectively optimizes the model training process, improves prediction performance, and provides reliable indicators for evaluating the prediction results.

## 4. Experimental Environment and Feature Extraction

### 4.1. Experimental Environment

In order to predict the electrical life of the AC circuit breaker, we referred to the relevant test conditions specified in the national standard GB14048.2 [26] and selected the NDM3-800 model AC circuit breaker manufactured by a certain electrical company as the test object. The experiment was conducted strictly according to the conditions specified in the national standard GB14048.2, and the specific parameter settings are detailed in Table 1.

### 4.2. Feature Extraction

The basic structure of a circuit breaker is illustrated in Figure 4.

The arcing between the stationary and moving contacts of an AC circuit breaker during opening and closing generates electrical erosion and welding, leading to the deterioration of contact conductivity and the residual electrical life of the circuit breaker. To precisely depict the deterioration state of an AC circuit breaker, it is necessary to collect the voltage and current waveforms of the breaker phases and extract feature parameters. The waveforms of normal interruption and closing in an AC circuit breaker can be analogized to Figure 5.

The specific state of the circuit breaker during the opening and closing process is shown in Table 2.

The circuit breaker’s feature parameters are computed by taking into account the contact state at each time point, as well as the sampling rate and the voltage and current information of the contacts, in accordance with the circuit breaker’s properties and operating principles. The time when the movable contact and the static contact first make contact after the coil is energized is defined as the closing time. The maximum voltage that the circuit breaker withstands during the closing process is defined as the maximum closing voltage. After the movable and static contacts first separate, due to the inertia and the magnetic field between the contacts, the movable contact may bounce back and forth for a short period of time until the arc is completely extinguished or the contacts are completely separated. This period is defined as the bounce time of the circuit breaker. After the movable and static contacts separate, the arc still exists in the circuit and continues to discharge. At this time, the arc is relatively stable, with small fluctuations in arc current and arc voltage. This period is defined as the arc plateau time. The time from the moment the movable and static contacts separate to the generation, extension, and extinction of the arc is defined as the arcing time. The arc burning energy refers to the heat generated by the arc from its generation to extinction. The arc drop count refers to the number of rapid voltage drops during the arcing period. It is generally considered that an arc drop occurs when the voltage rapidly drops by ≥50 V for a very short duration. Based on Figure 5 and Table 2, the equations for the feature parameters of the circuit breaker are shown in Table 3 [27].

The table includes the following variables: $U_{N}$ for the rated voltage, $u_{j}$ for the contact voltage, ${\;i}_{j}$ for the contact current, and N for the quantity of sampling points. $N_{w}$ represents the window size, $N_{h}$ represents the high-energy arcing count, and Δt represents the sampling time interval.

## 5. Case Analysis

The case study’s methodology is illustrated in Figure 6. The first step involves conducting a full life test of the AC circuit breaker in accordance with the specified requirements [26]. During this test, the voltage and current signals are recorded and used to extract the circuit breaker’s feature parameters. In the second step, the optimal subset of features is identified using PCA feature importance analysis and MIC correlation calculations. In the third step, the optimal feature subset is subjected to SG convolutional smoothing and then undergoes secondary feature extraction to obtain the optimal secondary feature subset. Next, the optimal secondary feature subset is normalized, and the residual electrical life of the AC circuit breaker is used as the prediction label. In the final step, the dataset is partitioned into separate training, validation, and testing subsets. These subsets are then fed into an LSTM model for training and evaluation. The prediction results are outputted after performing reverse normalization.

### 5.1. Feature Parameter Processing

The experiments were conducted using the NDM3-800 AC circuit breaker to verify the feasibility and accuracy of the proposed method. Figure 7 illustrates the comparison of the dynamic and static silver points of the three phases before and after the prototype testing.

The comparison shows that severe erosion of the dynamic and static silver points of the three phases occurs when the electrical life of the AC circuit breaker reaches its end. The original sequences and their variation trends of the prototype’s feature parameters are shown in Figure 8.

### 5.2. Feature Parameter Selection

Excessive feature parameters in AC circuit breakers can reflect the actual information of performance degradation. However, they may also introduce redundant information, leading to increased computational complexity and reduced accuracy in residual life prediction. Therefore, it is crucial to carefully select the extracted feature parameters [28].

In our research, we took into account the significance of feature parameters in relation to residual life and the interdependence between features. The best feature subset was selected using the PCA and MIC methods. Through the PCA feature importance analysis, we calculated the significance of feature parameters to residual life, as shown in Figure 9.

By using PCA, we can select the most representative and important feature subset, reducing the number of feature parameters and improving the performance and accuracy of the residual life prediction model. This selection process can eliminate redundant information, optimize feature selection, and provide more reliable inputs for subsequent prediction analysis.

Through the PCA importance analysis, we discovered that the average arc power had the least significance in relation to residual life. Furthermore, by calculating the correlation coefficients between the six feature parameters using the MIC, a heatmap of the correlation coefficients is displayed in Figure 10.

According to the results shown in the MIC heatmap, there is a strong correlation between the maximum closing voltage, arc platform time, and average arc power. Based on the analysis of feature importance and similarity between features, it can be concluded that retaining arc duration, arc platform time, and arc energy as the optimal subset of features is favorable.

This analysis indicates a high correlation between arc duration, arc platform time, arc energy, and the residual life. Therefore, in the process of feature selection, we choose to retain these features as the optimal subset, providing more reliable and effective input information. By filtering and selecting the optimal subset of features, we are able to decrease the number of features while still maintaining the most representative and essential features for predicting residual life. This helps optimize the model’s performance and improves the accuracy and reliability of residual life prediction.

### 5.3. Smoothing the Feature Sequence Based on the SG Algorithm

Based on the observations from Figure 8, it can be noticed that the original sequences of arc duration, arc plateau time, and arc energy exhibit significant fluctuations and noise. To reduce the impact of these fluctuations and noise, the Savitzky–Golay convolution smoothing algorithm was applied to smooth the feature sequences. Through the smoothing process, noise and outliers in the sequences can be filtered out, thereby extracting the main trends in the data. The specific results of the feature sequence before and after smoothing can be seen in Figure 11.

The application of the Savitzky–Golay convolution smoothing algorithm offers several important benefits. Firstly, it helps to remove unnecessary fluctuations and noise from the sequence, making the feature sequence smoother and more stable. This aids in revealing the true trends and patterns in the data, thereby improving the accuracy of prediction models. Secondly, through the convolution smoothing algorithm, we can preserve the important information in the sequence while maintaining the overall shape of the sequence. This ensures that the prediction model can capture the key features and trends in the feature sequence, thus enhancing prediction accuracy and reliability.

Specifically, by comparing the changes in the feature sequences before and after smoothing in Figure 11, it can be clearly observed that the smoothed sequences become smoother, more continuous, and exhibit reduced fluctuation amplitudes. This indicates that the convolution smoothing algorithm, Savitzky–Golay, effectively removes noise and outlier values, improving the reliability and stability of the feature sequence.

In conclusion, applying the Savitzky–Golay convolution smoothing algorithm to smooth the feature sequences such as arc duration, arc plateau time, and arc energy helps to reduce the influence of noise and outliers, extract the main trends in the sequence, and improve the accuracy and stability of prediction models. It provides robust support for achieving more accurate prediction results.

### 5.4. Model Parameter Settings

In this study, the Python 3.9 version and the Keras deep learning framework were used to build the model. The Graphviz module was utilized to visualize the model architecture parameters, as shown in Figure 12.

The input data are represented as a time series with a shape of (None, 1,5), where “None” indicates the batch size, “1” represents the time step, and “5” corresponds to the number of features in each time step.

To capture complex temporal dependencies and patterns in the data, we employ multiple LSTM layers in the architecture. The first LSTM layer (LSTM_1) takes the input time series and outputs a tensor with a shape of (None, 10, 265). The choice of 10 units in the output dimension is based on the characteristics of the problem, aiming to capture essential information from the time-series data while avoiding overfitting.

To introduce regularization and prevent overfitting, we utilize Dropout layers after LSTM_1 (Dropout_1) and LSTM_2 (Dropout_2). The Dropout layer with an output shape of (None, 10, 256) acts as a regularizer by randomly setting a fraction of the input units to zero during training, promoting model generalization and reducing the risk of overfitting.

Subsequently, we employ another LSTM layer (LSTM_3) to further refine the feature representation. The output shape of LSTM_3 is set to (None, 10, 128) to strike a balance between capturing relevant temporal patterns and reducing the model complexity.

LSTM_4 is the final LSTM layer in the architecture, with an input shape of (None, 10, 128) and an output shape of (None, 32). The 32 units in the output dimension are chosen based on an empirical trade-off between complexity and representation power.

The following Dense layer takes the output of LSTM_4 and outputs a tensor with a shape of (None, 1), which represents the final prediction for the remaining electrical life of the AC circuit breaker.

Lastly, the OutputLayer is used to match the desired output shape of (None, 1), ensuring compatibility with the model’s prediction and the actual target labels.

In summary, the network parameter settings are carefully chosen to accommodate the characteristics of the problem and the experimental dataset. The inclusion of multiple LSTM layers and Dropout layers aims to capture relevant temporal patterns, prevent overfitting, and strike an appropriate balance between model complexity and performance.

Given that failures of AC circuit breakers predominantly happen toward the end of their electrical life in practical scenarios, the dataset was split into training (80%), validation (15%), and testing (5%) sets. During the model training process, we employed the mean square error (MSE) as the loss function, and the loss function curve is shown in Figure 13.

In the graph, the blue line represents the training set, and the orange line represents the validation set. During the first few iterations (less than 40), both the training and validation sets exhibit a downward trend in the loss function. With an increase in the number of iterations, the loss function gradually reaches convergence. Notably, after 40 iterations, the loss function of the validation set remains stable without any significant increase, indicating that the model successfully avoids overfitting. Therefore, it can be concluded that the model performs well during the training process and has a good training effect.

### 5.5. Comparison of Prediction Results

Based on the comparison results in Figure 14 and Figure 15, the superiority of the proposed method can be demonstrated. In the graph, the orange line represents the predicted remaining electrical life, and the blue line represents the actual remaining electrical life. We present the remaining electrical life as a percentage by subtracting the number of operations performed by the circuit breaker under load from the total number of electrical life cycles and then dividing the result by the total number of electrical life cycles. This percentage value indicates the remaining percentage of the electrical life for the circuit breaker. The BP, GRU (gated recurrent unit), and LSTM models were compared before and after applying SG convolutional smoothing to the feature parameter sequences. The comparison results indicate that after SG convolutional smoothing, the BP, GRU, and LSTM models provide predictions of the remaining electrical life of the AC circuit breaker that are approaching the actual values. However, the LSTM neural network model with SG convolutional smoothing demonstrates the best match with the actual remaining electrical life curve and exhibits the smallest overall fluctuations. The specific quantitative analysis results can be seen in Table 4 and Table 5.

Based on the data presented in Table 4 and Table 5, the models with convolutional smoothing show improved prediction accuracy and reduced computation time compared to the models without SG convolutional smoothing. In the comparison before and after convolutional smoothing of the feature sequences, the LSTM method with SG convolutional smoothing exhibits lower prediction errors (RMSE, MAE, maximum error, and standard deviation of prediction accuracy) compared to the BP and GRU methods, indicating higher accuracy and stability. Moreover, the highest $R^{2}$ value suggests that the LSTM method has higher interpretability in predicting the remaining electrical life. Through feature sequence convolutional smoothing, the overall prediction accuracy of the LSTM method increases from 94.5% to 97.4%, demonstrating higher accuracy compared to the SG-smoothed BP and GRU methods.

To highlight the superiority of the proposed prediction method and compare the differences between different methods more clearly, the authors of this study conducted a comparative examination of the prediction error rates of each method on the test set after feature sequence convolutional smoothing, as shown in Figure 16. The comparison results, shown in Figure 17, demonstrate that the SG-smoothed LSTM method achieves comparable error rates to the BP and GRU methods while exhibiting overall higher stability. This indicates that the SG-LSTM method is more reliable for predicting the residual lifespan. In addition, the prediction results on the test set were contrasted with samples chosen at random.

The results of the comparison clearly show that the approach used in this research produces predictions that are closest to the true values and displays the greatest level of accuracy. In summary, the SG-LSTM-based residual electrical life prediction model for AC circuit breakers not only exhibits high levels of accuracy and robustness but also satisfies the demands of practical engineering applications. As such, it has considerable potential for use in predicting the residual electrical lifespan of low-voltage electrical equipment.

## 6. Conclusions

Based on the findings presented in this study, our proposed algorithm demonstrates commendable accuracy, achieving an encouraging 97.4% prediction rate for the remaining electrical life of AC circuit breakers. Through a thorough case analysis, we have observed relative advantages of our method compared to widely used prediction models, such as BP and GRU.

Moreover, our research indicates that the application of this algorithm extends beyond traditional planned maintenance, potentially enabling more efficient state-based maintenance practices in power systems. Additionally, our method shows promising potential for implementation in various scenarios, facilitating predictive maintenance for electrical equipment and optimizing resource allocation.

Looking to the future, we recognize certain challenges and limitations in the practical implementation of our proposed method. Collaboration with industry experts and continuous research will be essential in addressing specific operational and environmental factors that may impact prediction accuracy. Ensuring the long-term viability of our predictive models in real-world applications relies on continuous data collection and model refinement.

In conclusion, this study provides an analytical framework for enhancing residual life prediction algorithms and investigating power supply reliability. Moving forward, we will conduct further research to explore the model’s robustness and scalability, ensuring its effectiveness across various operational environments and load conditions in real-world power systems.

## Figures and Tables

**Figure 1 sensors-23-06860-f001:**
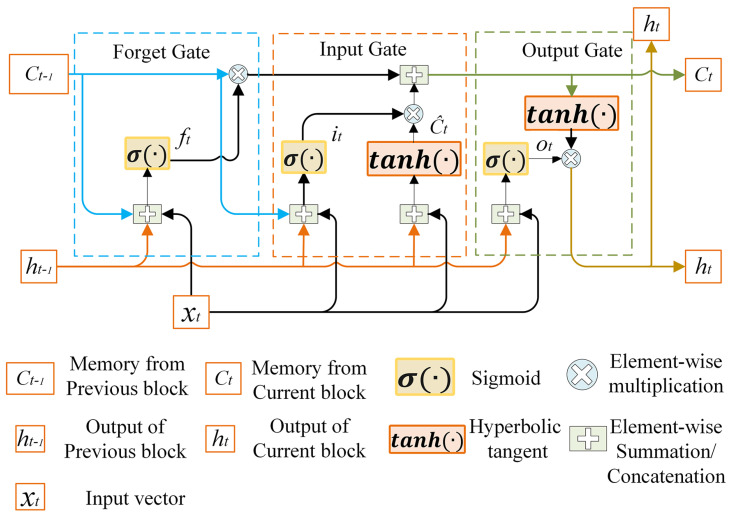
Internal structure diagram of LSTM.

**Figure 2 sensors-23-06860-f002:**
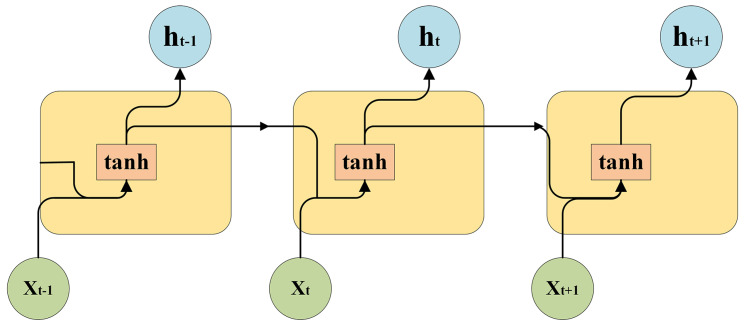
Circular structure of LSTM.

**Figure 3 sensors-23-06860-f003:**
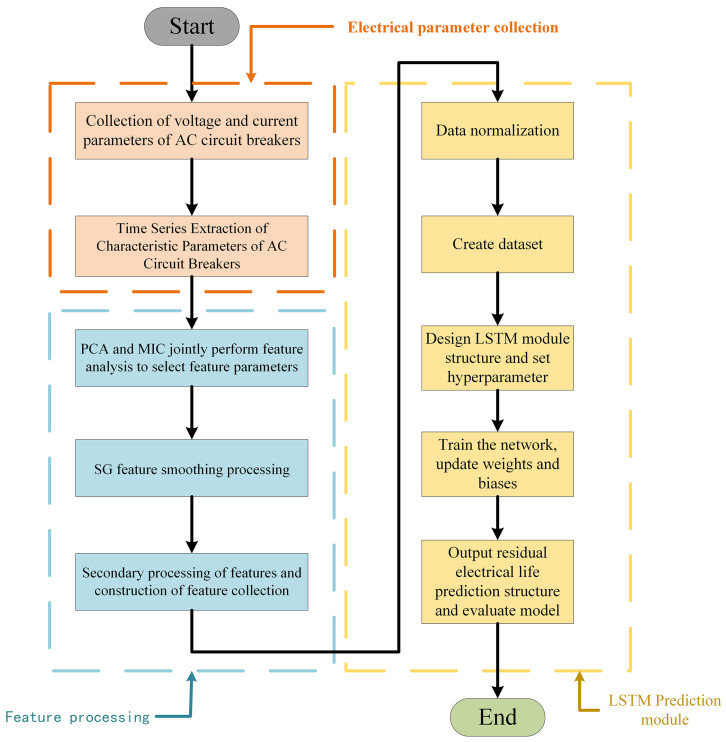
Schematic diagram of the model structure.

**Figure 4 sensors-23-06860-f004:**
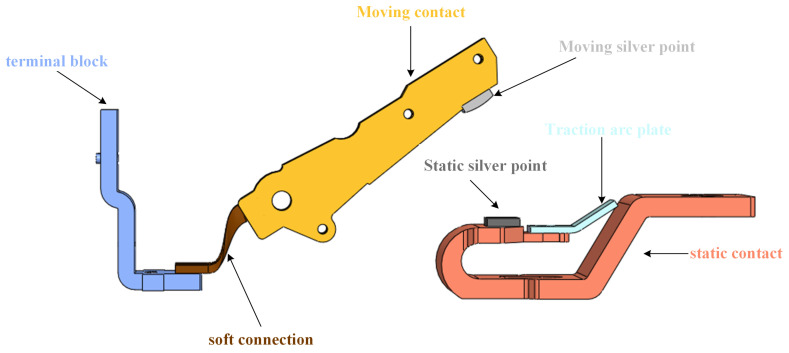
Structure diagram of AC circuit breaker.

**Figure 5 sensors-23-06860-f005:**
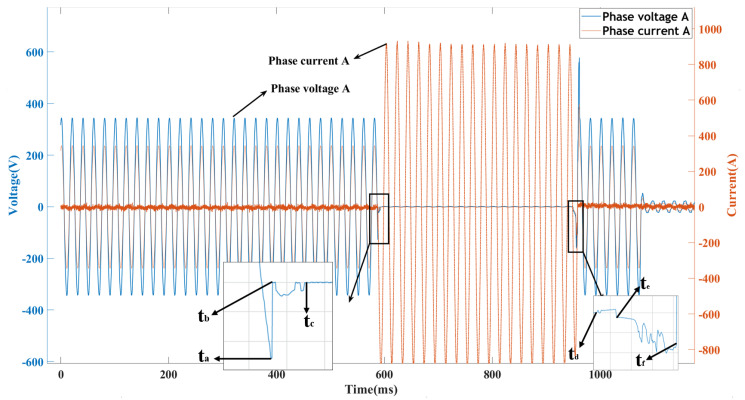
The normal breaking and closing waveform.

**Figure 6 sensors-23-06860-f006:**
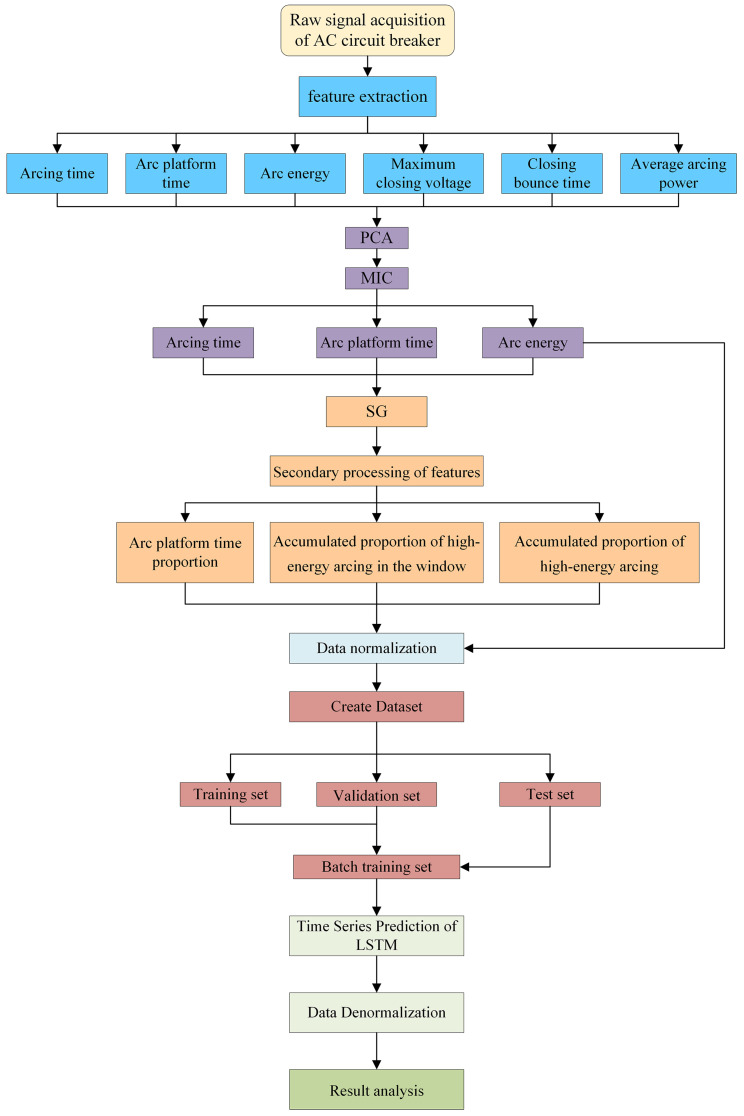
Flow chart of case analysis.

**Figure 7 sensors-23-06860-f007:**
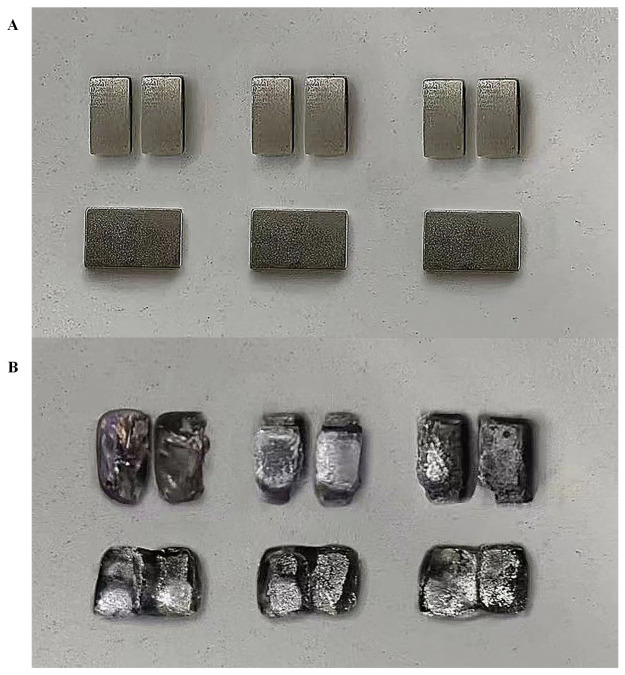
Comparison of dynamic and static silver point tests before and after. (**A**) Static silver point tests before and after. (**B**) After the test.

**Figure 8 sensors-23-06860-f008:**
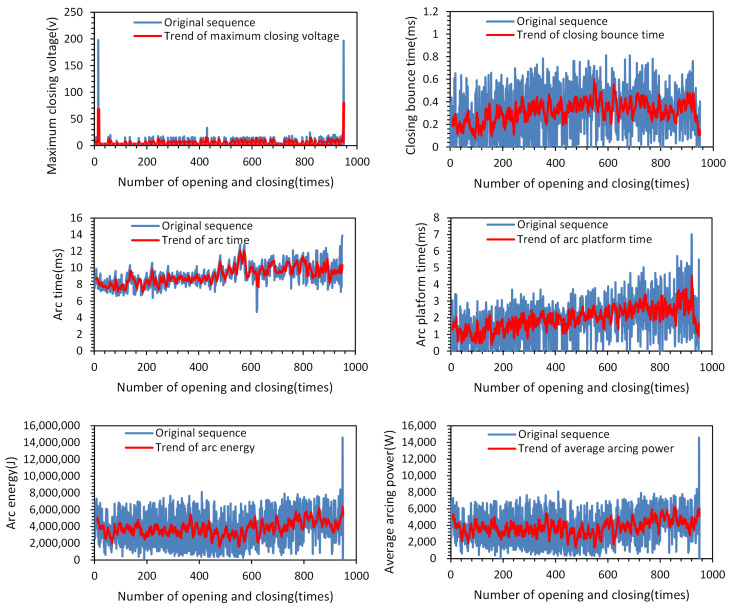
Characteristic trend chart.

**Figure 9 sensors-23-06860-f009:**
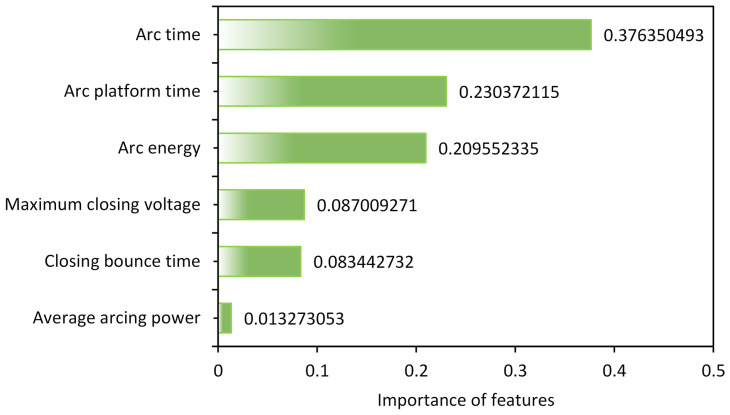
Comparison of PCA feature importance analysis.

**Figure 10 sensors-23-06860-f010:**
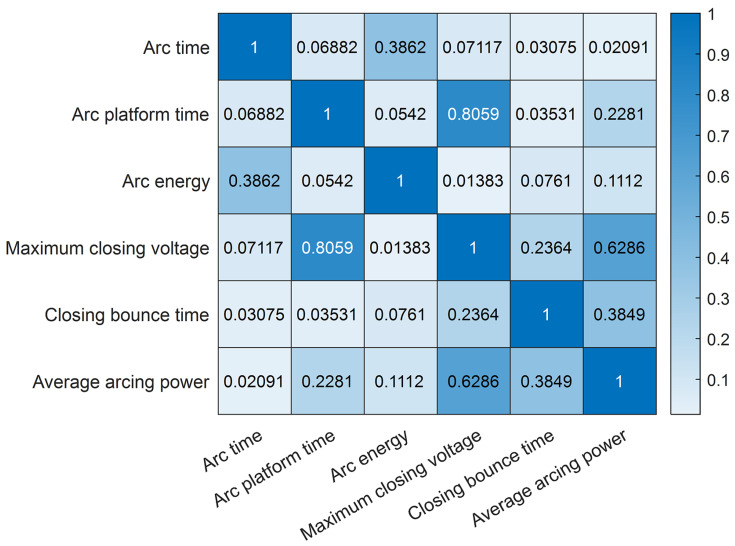
Heatmap of correlation coefficients.

**Figure 11 sensors-23-06860-f011:**
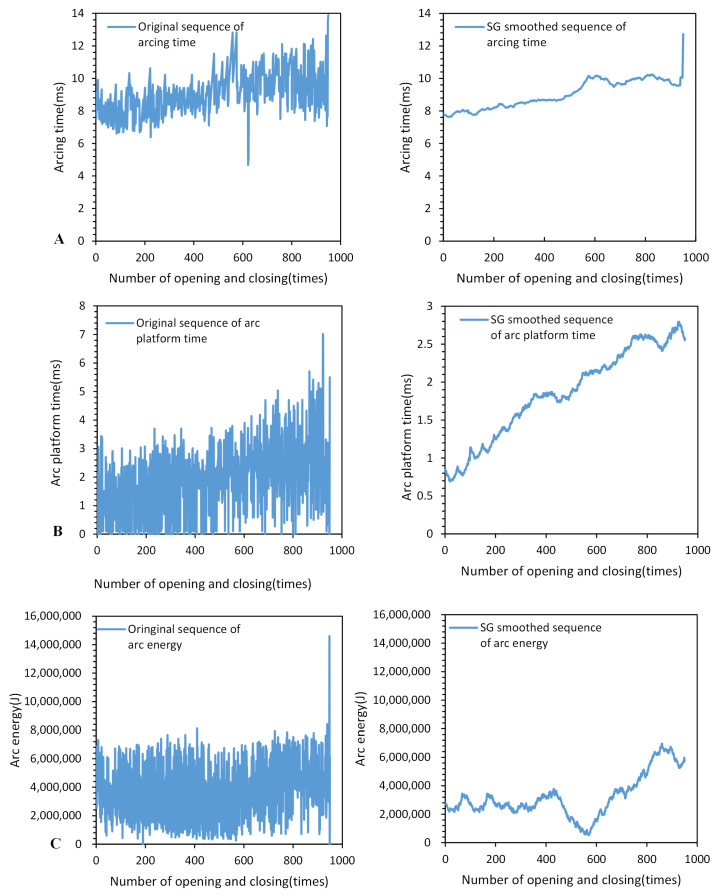
Comparison of feature sequence before and after smoothing: (**A**) Comparison of arcing time series before and after smoothing. (**B**) Comparison of arc platform time series before and after smoothing. (**C**) Comparison of arc energy time series before and after smoothing.

**Figure 12 sensors-23-06860-f012:**
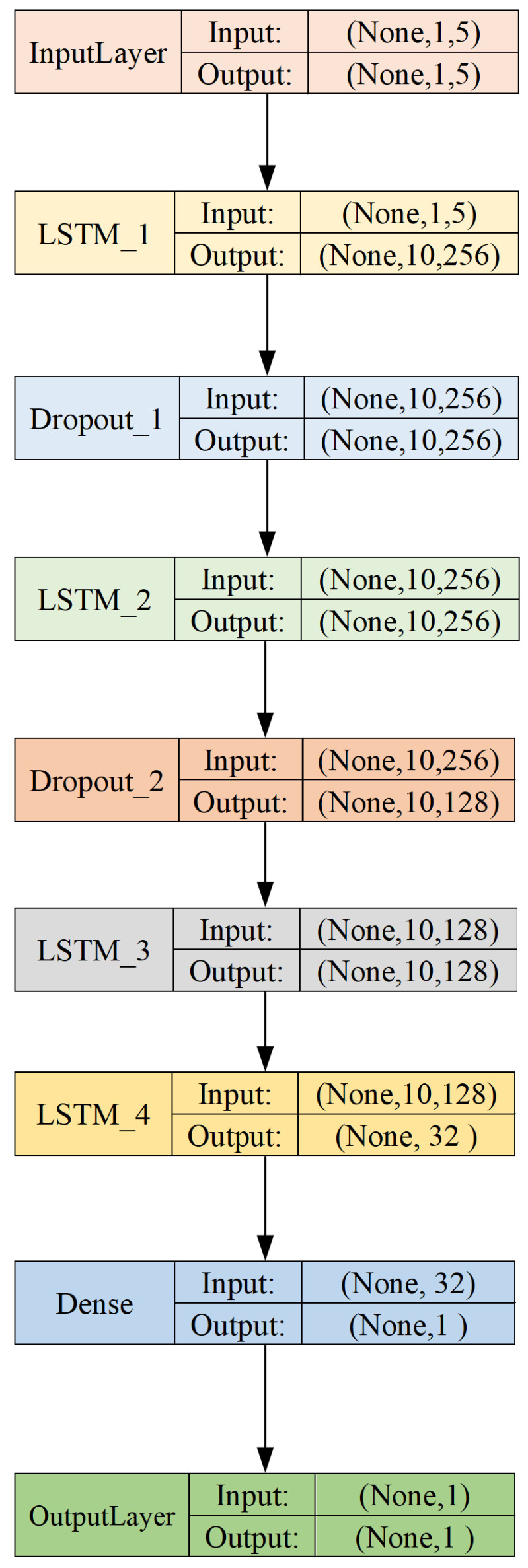
Structural parameters of the model.

**Figure 13 sensors-23-06860-f013:**
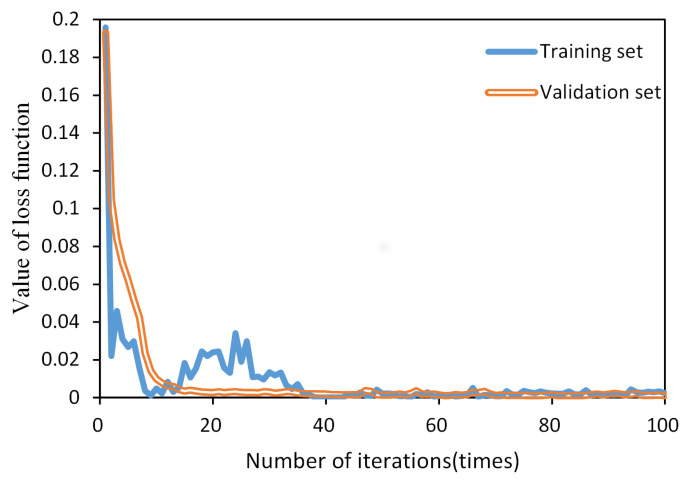
The loss function curves.

**Figure 14 sensors-23-06860-f014:**
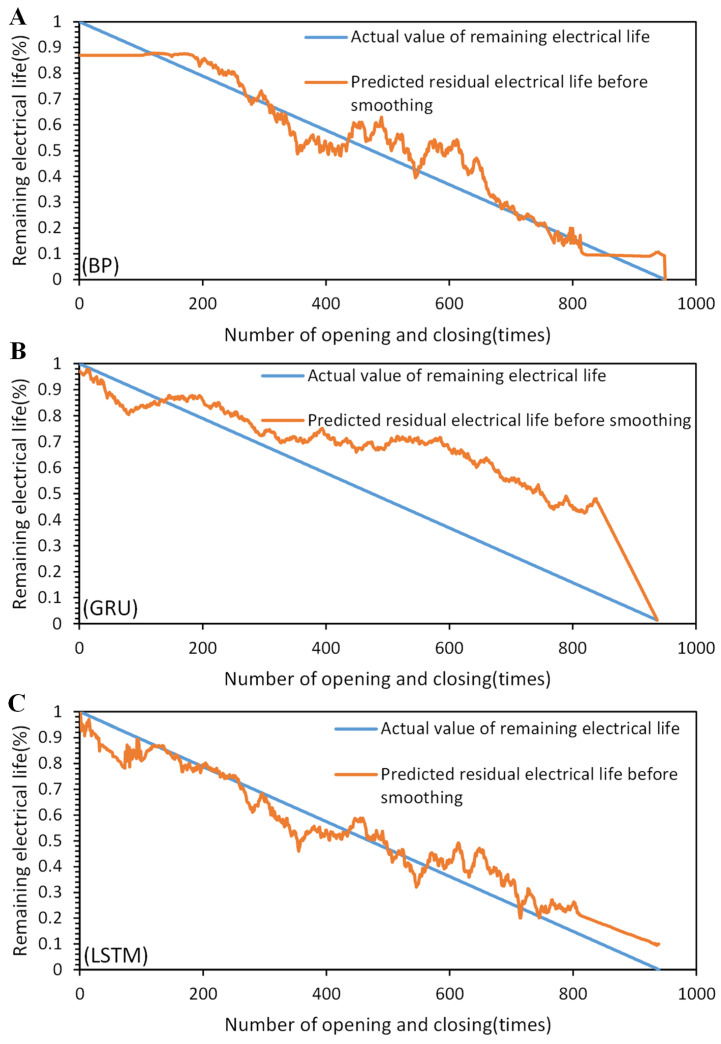
Comparison of prediction results before SG smoothing: (**A**) BP. (**B**) GRU. (**C**) LSTM.

**Figure 15 sensors-23-06860-f015:**
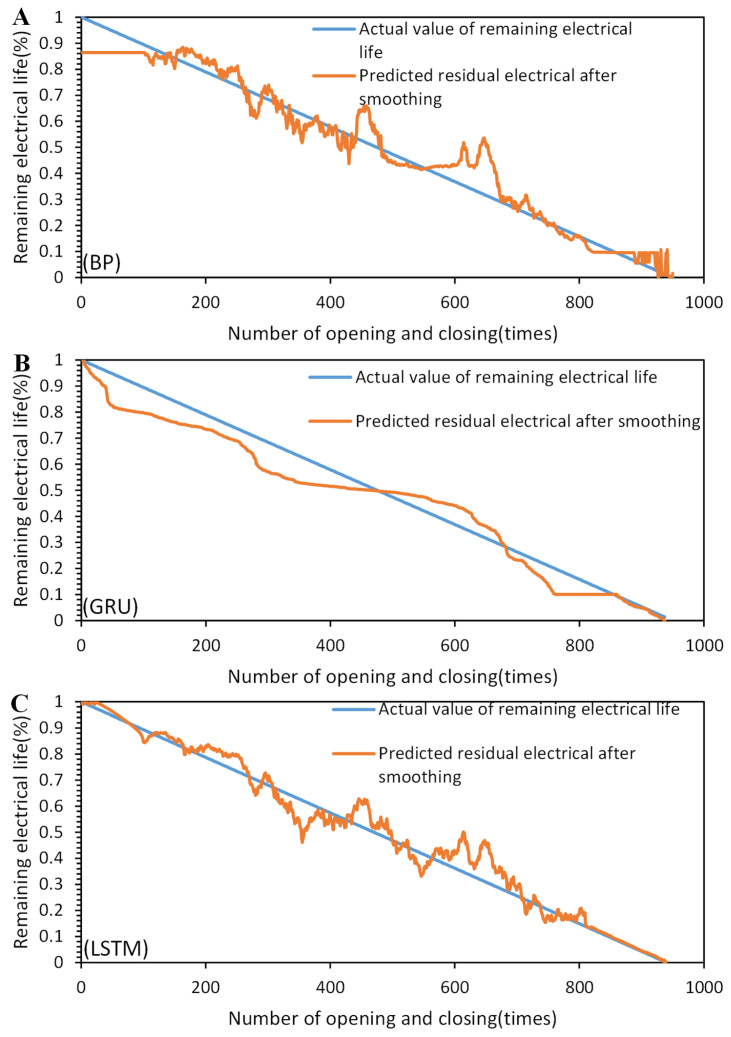
Comparison of prediction results after SG smoothing: (**A**) BP. (**B**) GRU. (**C**) LSTM.

**Figure 16 sensors-23-06860-f016:**
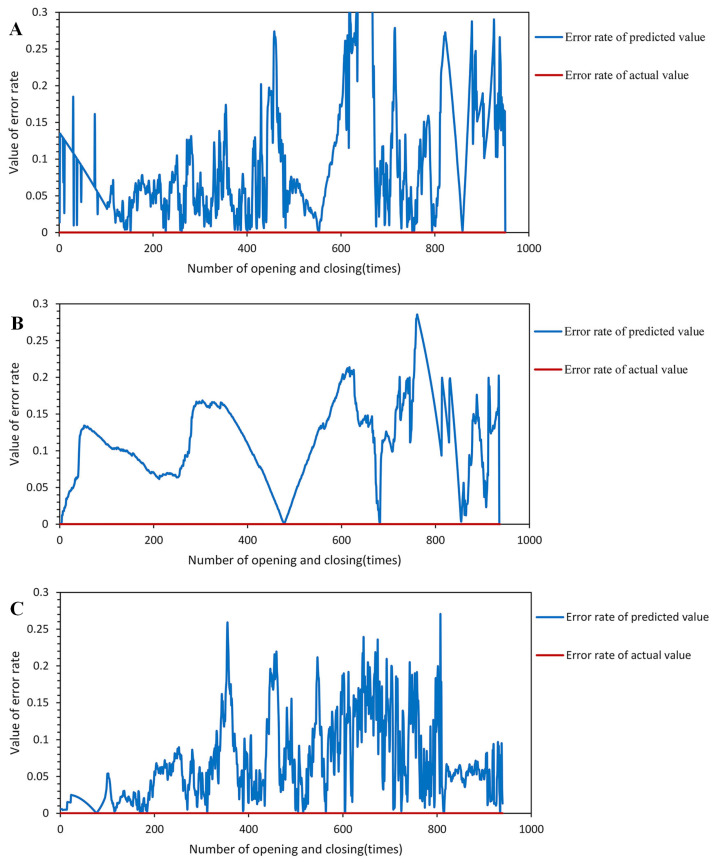
Comparison of model errors: (**A**) BP. (**B**) GRU. (**C**) LSTM.

**Figure 17 sensors-23-06860-f017:**
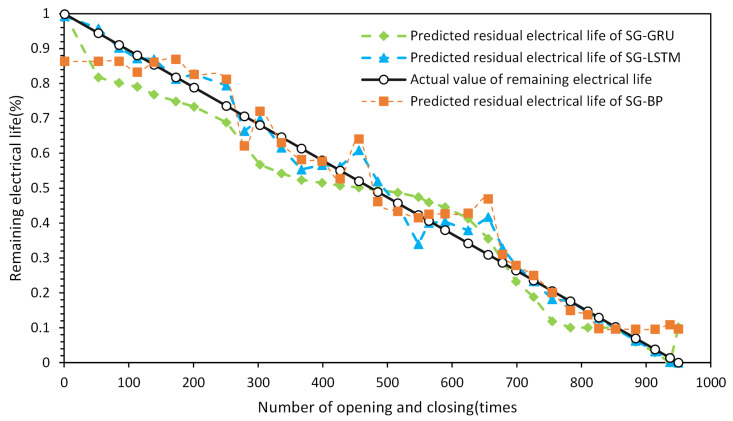
Sampling comparison diagram of prediction results.

**Table 1 sensors-23-06860-t001:** Test parameters of GB14048.2.

Test Conditions	GB14048.2
Load voltage	(1 + 5%)AC415V
Load current	800 A
Power factor	0.8
Operation frequency	20 times/h
Test frequency	50 Hz

**Table 2 sensors-23-06860-t002:** Status of Contacts.

Time	Status
$t_{a}$	The rotation of the mechanism drives the movement of the movable contact.
$t_{b}$	The first closure of the movable and stationary contacts generates an electric current.
$t_{c}$	The contacts remain stably closed, and the contact voltage stabilizes.
$t_{d}$	The mechanism rotates in the opposite direction, causing the contacts to separate, resulting in an arc between the contacts.
$t_{e}$	After the contacts separate, the arc continues to discharge between the contacts.
$t_{f}$	The arc extinguishes when the current crosses.

**Table 3 sensors-23-06860-t003:** Definition of feature parameters.

Serial Number	Feature Name	Computational Formulas
1	Maximum closing voltage $U_{\max}$	$U_{\max}=\frac{2}{\pi }\sqrt{2}U_{N}$
2	Closing bounce time $t_{\mathrm{bounce}}$	$t_{\mathrm{bounce}}=t_{c}-t_{b}$
3	Arc platform time $t_{\mathrm{arc}\_\mathrm{platform}}$	$t_{\mathrm{arc}\_\mathrm{platform}}=t_{d}-t_{e}$
4	Arc time $t_{\mathrm{arc}}$	$t_{\mathrm{arc}}=t_{f}-t_{d}$
5	Arc energy E	$E=\sum_{j=d}^{f}u_{j}\cdot i_{j}\cdot \Delta t$
6	Average arcing power P	$P=\frac{\sum_{j=d}^{f}u_{j}\cdot i_{j}}{N}$
7	Arc platform time proportion $r_{\mathrm{arc}}$	$r_{\mathrm{arc}}=\frac{t_{\mathrm{arc}\_\mathrm{platform}}}{t_{\mathrm{arc}}}$
8	Accumulated proportion of high-energy arcing in the window $r_{w}$	$r_{w}=\frac{\sum_{d}^{d-N_{w}}N_{h}}{N_{w}}$
9	Accumulated proportion of high-energy arcing $r_{h}$	$r_{h}=\frac{\sum_{d}^{f}N_{h}}{N}$

**Table 4 sensors-23-06860-t004:** Model evaluation indicators before SG smoothing.

Prediction Models	BP	GRU	LSTM
RMSE	0.067	0.201	0.068
MAE	0.054	0.169	0.057
$R^{2}$	0.946	0.500	0.945
Max error	0.187	0.363	0.162
Effective precision	0.946	0.500	0.945
Standard deviation of prediction accuracy	0.064	0.131	0.067
Calculation time	55.6 s	47.3 s	42.5 s

**Table 5 sensors-23-06860-t005:** Model evaluation indicators before SG smoothing.

Prediction Models	BP	GRU	LSTM
RMSE	0.061	0.066	0.046
MAE	0.046	0.056	0.033
$R^{2}$	0.955	0.947	0.974
Max error	0.217	0.127	0.116
Effective precision	0.955	0.947	0.974
Standard deviation of prediction accuracy	0.060	0.054	0.045
Calculation time	52.3 s	45.9 s	40.3 s

## Data Availability

The raw data supporting the conclusion of this article will be made available by the authors, without undue reservation.

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
