# Peer review of "The Prediction of Residual Electrical Life in Alternating Current Circuit Breakers Based on Savitzky-Golay-Long Short-Term"

_sensors, 2023, doi:10.3390/s23156860_

Round 1
Reviewer 1 Report
It is important to predict the remaining electrical life of AC circuit breakers and replace the circuit breaker with latent defect for the maintenance of low voltage AC network.
The authors propose a method based on Savitzky-Golay convolution smoothing and long short-term memory neural network for the electrical life prediction of AC circuit breakers. In addition, principal component analysis and maximum information coefficient are used to remove redundancy in the feature parameters. The feasibility and effectiveness of SG-LSTM method has been demonstrated by numerical experiment. The test setup and condition of experiments on NDM3-800 AC circuit breaker should be described. Besides, the quality of images in this paper should be improved.
Reviewer 2 Report
Dear Authors,
the paper addresses and interesting point, as it is applicable to AC circuit breakers in this case, but also to other devices and problems of determination of operating life. However, the paper is quite confused and does not explain the methods: how you used an algorithm, how you tested it is correctly configured, the learning / testing for the neural network, the process flow for learning, prediction, verification.
1) Line 83-85. Sentence has lost the subject.
2) i) Sections are not numbered correctly as they start from 0. Also formatting of section headings is a problem.
ii) Actual section 1 (principles) and 2 (prediction model) are mixed. You speak of PCA , recurrent neural networks, then some metrics for error, but a lot of points are not clear, such as (a) the sequence of application of PCA, neural network and metrics, (b) how the learning of the nn is carried out, (c) which experimental data are used for the learning and for the verification of the acquired learnt space.
First, this must be clearly explained with criteria to judge a good or bad learning and in general performance. Then, you can apply to a specific case.
3) Line 153: metric not metrics
4) Eq (12)-(14). To clarify which kind of operation between matrices is represented by the thick dot.
Be careful that the same symbol is used for something different in eq (15)-(18).
And in eq (19) you use an asterisk.
So, multiplications must be revised and receive a uniform and adequate notation.
5) Line 330. Confusion for the selection of features that are not so well explained. Sampling rate is not a feature and has little to do in general.
6) Section 4.2 and figure 10. You speak of "correlation coefficients" but you should make explicit reference to your previous formulas, or explain here in detail how such coefficients are calculated.
7) Section 4.4. You need to discuss how you set up such parameters, namely what are the criteria with respect to the characteristics of the problem and of the experimental data set.
8) Figure 13. Which loss function? which training and validation set and phase? how this was carried out?
9) GRU not explained. A lot of acronyms must be verified and explained when they are introduced.
10) Section 4.5. The "actual value of remaining electrical life" in the figures must be introduced and explained: how it is determined? what is its uncertainty as it is used as reference? why it has a linear behavior?
11) Conclusions must explain the quality of the results, the most important findings and any future development, not how you did the experiments, or how to feed the data from one algorithm to the other. Another important point (that is part of the discussion) is the comparison with past published results.
12) References not in MDPI style.
Ref [18] evidently wrong.
Be careful to lost capital letters in formatting of references: e.g. ref [8] "pv" is "PV", ref [14] "nature" is "Nature".
DOIs are mostly missing.
see comments above
Reviewer 3 Report
Dear Authors,
I have some comments on your article:
1. Literature should be checked if there are no newer items. Especially from the last 18 months.
2. All indexes in symbols in text and equations should be checked carefully.
3. In the Conclusions section, please write something more about the real possibility of implementing the proposed methods of AC circuit breaker remaining electrical life prediction.
Round 2
Reviewer 2 Report
Dear Authors,
thank you for your replies and amendments to the manuscript.
General check for improvement of English form.